# siRNA Functionalized Lipid Nanoparticles (LNPs) in Management of Diseases

**DOI:** 10.3390/pharmaceutics14112520

**Published:** 2022-11-19

**Authors:** Tutu Kalita, Saba Abbasi Dezfouli, Lalit M. Pandey, Hasan Uludag

**Affiliations:** 1Bio-Interface & Environmental Engineering Lab, Department of Biosciences and Bioengineering, Indian Institute of Technology Guwahati, Guwahati 781039, India; 2Department of Chemical & Materials Engineering, Faculty of Engineering, University of Alberta, Edmonton, AB T6G 2V2, Canada; 3Faculty of Pharmacy and Pharmaceutical Sciences, University of Alberta, Edmonton, AB T6G 2V2, Canada; 4Department of Biomedical Engineering, Faculty of Medicine and Dentistry, University of Alberta, Edmonton, AB T6G 2V2, Canada

**Keywords:** nanotechnology, lipid nanoparticles, siRNA, surface modification, cancer, hepatitis B, COVID-19

## Abstract

RNAi (RNA interference)-based technology is emerging as a versatile tool which has been widely utilized in the treatment of various diseases. siRNA can alter gene expression by binding to the target mRNA and thereby inhibiting its translation. This remarkable potential of siRNA makes it a useful candidate, and it has been successively used in the treatment of diseases, including cancer. However, certain properties of siRNA such as its large size and susceptibility to degradation by RNases are major drawbacks of using this technology at the broader scale. To overcome these challenges, there is a requirement for versatile tools for safe and efficient delivery of siRNA to its target site. Lipid nanoparticles (LNPs) have been extensively explored to this end, and this paper reviews different types of LNPs, namely liposomes, solid lipid NPs, nanostructured lipid carriers, and nanoemulsions, to highlight this delivery mode. The materials and methods of preparation of the LNPs have been described here, and pertinent physicochemical properties such as particle size, surface charge, surface modifications, and PEGylation in enhancing the delivery performance (stability and specificity) have been summarized. We have discussed in detail various challenges facing LNPs and various strategies to overcome biological barriers to undertake the safe delivery of siRNA to a target site. We additionally highlighted representative therapeutic applications of LNP formulations with siRNA that may offer unique therapeutic benefits in such wide areas as acute myeloid leukaemia, breast cancer, liver disease, hepatitis B and COVID-19 as recent examples.

## 1. Introduction

RNA interference (RNAi) is a cellular process that occurs in nature. It plays a major role in silencing a specific gene of the target either by endogenous microRNAs or by intracellularly delivering the small interfering RNAs (siRNAs). The RNAi process involves binding of the mediators to the ’UTR region of mRNA, blocking the translation of mRNA into protein, leading to either the gain or loss in function depending on the protein function. In addition to the regulation of gene expression [1], RNAi is also involved in modulating cellular defence mechanisms against various viral pathogens [2]. Therefore, employing this mighty technology could be useful in a better understanding of gene signalling and functionality even at a whole-genome level [1,3,4]. In the past two decades, numerous papers have been published highlighting its potential for the treatment of a wide range of diseases such as cancer, autoimmune disease, dominant genetic disorders, and infections caused by the virus [5,6,7,8,9,10,11]. However, it is well recognized that uptake of the naked form of nucleic acid inside the cell is very poor owing to its larger molecular size and a highly anionic backbone arising from the presence of phosphate groups [12,13]. In addition, small RNAs are vulnerable to enzymatic degradation and susceptible to renal clearance. The siRNAs are very fragile and immunogenic in comparison to their oligonucleotide counterparts [14,15,16]. As a consequence, the direct delivery of naked nucleic acid is avoided, and the delivery of siRNAs remains a challenge for researchers. To overcome this problem, chemical modification at various sites of the siRNA backbone such as phosphate, nucleobase, ribose, and terminal conjugate groups lead to the improvement in serum and thermal stability in vitro conditions together with gene silencing activity was observed [17,18]. However, when applied under in vivo conditions, modified siRNA showed mixed efficacy, leading to decreasing therapeutic effect [1]. Therefore, there is a requirement for the development of an adequate delivering system to aid the proper cellular uptake of siRNA, which is safe, stable, and target-specific, in order to achieve successful clinical applications.

Nanotechnology-derived nanocarriers have proven to be a powerful strategy for overcoming the challenges faced by conventional delivery systems. Nanocarriers have motivated the attention of researchers in delivering various drugs, proteins, oligonucleotides, and genes to the targeted sites of interest [19,20,21,22]. Among the various nanocarriers, lipid NPs (LNPs) have attracted the attention of researchers since lipidic NPs are by and large considered non-toxic, biocompatible, and are mainly formulated using physiological lipids [23,24]. Using lipidic NPs as a carrier is beneficial in terms of enhanced physical stability, and a low level of toxicity [25]. LNPs can also be rapidly produced at a larger level, which can be beneficial for clinical trials and meeting commercial demand [26]. The delivery of RNAi using LNP as a delivery vehicle could enhance its long-term shelf life in serum and be capable of researching to its target site of interest. Additionally, the usage of LNPs minimizes the cost of developing chemically altered siRNA. Studies showed that the inside the cell, LNPs become internalized through the process of endocytosis and thereby could favour endosomal escape. The LNP-mediated intracellular delivery of siRNA helps to bypass endosomal degradation using a process of lipid mixing and assuming the inverted hexagonal structures [1]. The schematic representation of action of LNP-mediated siRNA delivery into the cell is shown in Figure 1. The physical properties of LNPs such as size, surface charge distribution, and chemical properties such as types of lipids used and its composition, the nitrogen/phosphate (N/P) molar ratio, and RNA encapsulation efficiency, are some of the characteristics of LNPs that govern the overall biodistribution and therapeutic effects. Thus, the aforementioned properties of LNPs need to be optimized to maximize the therapeutic effects [27].

The present review focuses on the role of LNPs as the delivery agent of siRNA applied in various applications. We have provided a brief introduction to the types and properties of LNPs and reviewed the current usage of LNPs for siRNA therapy. Surface modification of LNPs with targeting ligands such as antibodies, antibody fragments, and peptides has been discussed. Biological barriers during the delivery of LNPs including protection against nucleases, evading the mononuclear phagocytic system (MPS), renal glomerular filtration, and systematic administration have been discussed in detail. The potential applications of siRNA-functionalized LNPs for cancer, liver disease, hepatitis-B and COVID-19 have been summarized along with recent examples. We additionally highlighted the computational studies for designing siRNAs which could efficiently target SARS-CoV-2 virus and could be successfully applied for development of siRNA-based therapies in a very short span of time, thereby reducing the risk of virus outbreak in the future.

## 2. Overview of LNPs

Across the pharmaceutical industry, LNPs are rapidly emerging as a promising carrier capable of delivering a wide range of therapeutic agents. As a potential carrier, the applications of LNPs are not only confined to the clinical therapeutic field but have also been extended to other fields, such as nutrition, medical imaging, cosmetics, agriculture, as well as nanoreactors [29]. Several types LNPs have been developed (Figure 1 and Figure 2) using a large number of components (Table 1) with a variety of fabrication techniques (Table 2).

### 2.1. Liposomes

In 1961, Alec D. Bangham, a British haematologist, described liposomes as the first types of LNPs. Liposomes are spherical vesicular structures of 10–1000 nm in size. They are mainly composed of one or more phospholipid bilayers which are prepared using different types of naturally occurring phospholipids and cholesterol [31,32]. Phospholipids could self-assemble into liposomes having an aqueous core surrounded by lipid bilayers. This structure allows efficient encapsulation of hydrophilic drugs in the core, while hydrophobic drugs are sandwiched in the external bilayers. Phospholipids consist of different hydrocarbon structures which include phosphatidic acid (PA), phosphatidylethanolamine (PE), phosphatidylglycerol (PG), phosphatidylserine (PS) and phosphatidylcholine (PC). All these phospholipids share common amphiphilic structures having a polar head group with two hydrophobic alkyl tails. Under physiological pH, the polar head group provides the lipids with anionic (PA, PG, and PS) or neutral (PE and PC) moieties, while the presence of alkenes tails helps in lowering the solid-to-liquid transition temperatures. This property of phospholipids promotes the formation of liposomes at ambient temperature using unsaturated lipids [33]. The gaps between the phospholipids are mainly filled with the incorporation of cholesterol, and this helps in stabilizing the lipid bilayer. In addition, cholesterol enhances the stability of liposomes in the presence of serum proteins and promotes membrane fusion. Together with PC, the inclusion of cholesterol results in the formation of stable lipid bilayers and thus has been commonly used for LNP formulations related to siRNA delivery [34,35].

### 2.2. Solid Lipid NPs (SLNs)

In 1991, SLN was synthesized as an alternative to liposomes. SLNs are colloidal carriers of the nano-size in the range of 40–1000 nm. SLNs possess a core of solid lipids in which a bioactive component is embedded. The core is stabilized by using a surfactant coating [36], which also helps in reducing the interfacial tension which occurs between the aqueous phase and lipid during the formulation of SLNs. The surfactants present at the interface of the lipid and aqueous phase tend to adsorb as a flexible and mechanically strong monolayer and, thus, improve the physical stability of the nanodispersion during the process of manufacturing and storage. Studies showed that with the increasing amount of surfactant added during the formulation, the size of the SLNs decreases. For example, Kheradmandnia et al. reported that the increasing concentration of surfactant mixtures (tween 80 and egg lecithin) from 0.5 to 1.5% (*w/v*) resulted in the decreased mean size of SLNs (1:1 mixture of carnauba wax and beeswax) from 100 to 65 nm [31]. The low-molecular-weight surfactants tend to take less time for redistribution between the surfaces of the prepared particles and micelles, while a longer time is taken by the high-molecularweight surfactants for redistribution [37]. Ebrahimi et al. used stearic acid and glyceryl mono stearate (GMS) as lipid components to prepare SLN. The results showed that using pluronic, Tween 80, and lecithin-based SLNs resulted in the formation of particles of smaller sizes than polyvinylpyrrolidine (PVP) and polyvinyl alcohol (PVA)-based SLNs [38]. Therefore, the correct choice of surfactant is very crucial for the proper preparation of SLNs [39]. Though the SLN has several potential applications in research and drug delivery, however, it has some limitations such as low storage stability, reduced drug release during storage, and reduced drug loading capacity due to the lower solubility of some drugs in the solid lipid [40,41].

### 2.3. Nanostructured Lipid Carriers (NLCs)

In order to overcome some limitations associated with SLNs, NLCs were developed and are considered as the second generation of LNPs [42]. NLCs, together with solid lipids, also make use of liquid lipids, and this helps to reduce the crystallinity of the lipid matrix during NP formation [43]. As a result of reduced crystallinity, the expulsion of the drug from the matrix is suppressed and thus improves the drug-loading capacities of the carrier [44]. In addition, NLCs offer a higher loading capacity for a variety of active compounds with a lower water content of the particle suspension [45]. Typically, the particle size of NLCs lies between 150 and 300 nm, although carrier size smaller than 40 nm and larger than 1000 nm can also be developed as per requirements. The major drawback of NLCs is the optimization process that requires an appropriate ratio of solid/liquid lipids to form stable NPs [46].

### 2.4. Nanoemulsions (NEs)

NEs are a colloidal dispersion comprising of two immiscible liquids such as oil and water which are stabilized using a surfactant/cosurfactant. NEs are either transparent or translucent and consist of fine dispersions of nano-sized droplets ranging from 1 to 200 nm [47]. Typically, NEs as a result of their smaller size possess a greater surface-to-volume ratio making it more stable and permeable. NEs are classified as biphasic (water-in-oil [W/O] or oil-in-water [O/W]) or maybe as triphasic structures (oil-in-water-in-oil [O/W/O] or water-in-oil-in-water [W/O/W]) [48]. This allows the transport of both hydrophilic and lipophilic molecules depending upon the types of surfactant and proportions of the different phases used. The application of NEs as a drug carrier offers several advantages such as the pronounced small size, kinetic stability, and biodegradable in nature, prevents the susceptible drug from hydrolysis, and thus protects it from enzymatic degradation. In addition, NEs can be applied for different routes of administration such as oral, nasal, intravenous, topical, pulmonary, transdermal, and ocular routes, and hence are helpful in delivering drugs both locally or systemically [49,50,51]. The major drawback of NEs is the requirement of a higher surfactant concentration and the possibility of phase separation.

**Table 1 pharmaceutics-14-02520-t001:** List of lipids and surfactants used for the synthesis of LNPs [1,52,53,54,55].

Solid Lipids	Liquid Lipids	Common Lipids	Surfactants
Paraffin Tricaprin Trilaurin Trimyristin Tripalmitin Tristearin Acyl glycerols Glyceryl behenate Glyceryl distearate Glyceryl monostearate Glyceryl monooleate Glyceryl palmitostearate Cetyl palmitate Beeswaxes Palmitic acid Stearic acid Behenic acid Decanoic acid	Lutrol^®^ F68 Miglyol^®^ 812 Castor oil Oleic acid	Phospholipids PA; PCPE; PG; PS Ionizable cationic lipid DODAP; DLin-K-DMA; DLinDMA; DlinMC3-DMA; DLin-KC2-DMA Additional lipids Cholesterol; DMG-PEG_2000_; DSPE-PEG_2000_PE	Lecithin Polysorbate 80 Polysorbate 60 Polysorbate 20 Poloxamer 407 Poloxamer 188 Sodium oleate Sodium dodecyl sulphate Polyvinyl alcohol Butanol Butyric acid

**Table 2 pharmaceutics-14-02520-t002:** Fabrication methods used for the preparation of lipid-based NPs.

Procedure	Advantages	Disadvantages
Hot Homogenization		
It is carried out at temperatures greater than the melting point of solid lipids. The drug and lipids are melted together and added in a hot aqueous phase having the surfactants, using a high-shear mixing device. The system is then cooled leading to the solidification of lipids and the formation of NPs [56]	Simple and solvent-free technique Easy to scale up Short production time Small size particle	Not suitable for thermosensitive drug May lead to penetration of drug into the aqueous phase during the homogenization process Coexistence of supercooled melts, various colloidal structures during lipid crystallization
Cold Homogenization		
Drug is dissolved in the melted lipid mixture and the mixture is quickly cooled down using dry ice or liquid nitrogen and solidified. It is then grinded into a very fine powder using high-pressure milling. The resulting microparticles are dispersed in a cold aqueous phase having the surfactant [57]	Applicable for temperature-labile drugs or hydrophilic drugs Avoid high-temperature treatment Coexistence of other colloidal structures is minimum	Requirement of microsized drug particles in dispersion before homogenization step
Solvent Emulsification Evaporation		
The lipid is first dissolved in a non-polar organic solvent and then emulsified by high-speed homogenization in an aqueous phase. The solvent is evaporated using mechanical stirring under reduced pressure and room temperature, resulting in the formation of LNPs [58]	Simple procedure Useful for thermolabile drugs as no heat is required during production	The insolubility of lipids in organic solvents. Presence of solvent residues Thermodynamically unstable
Ultrasonication		
Applies the temperatures that are greater than the melting point of the solid lipid. The melted lipid is then dispersed into the warm aqueous phase containing the surfactant. The pre-emulsion is then placed into the ice-water bath and subjected to ultrasonication using a probe sonicator [59]	Low energy requirement	The inability to produce NPs of narrow size distribution, thereby leading to instability during storage, Chances of metallic contamination of the product
Supercritical Fluid extraction of Emulsion		
An aqueous solution containing lipid, drug, and surfactant is placed in a high-pressure homogenizer to form an oil/water emulsion. A supercritical fluid such as CO_2_ is used for the removal of the solvent from o/w emulsions after which lipid NPs are obtained [60]	Solvent free method Dry powdered product Wide range of miscibility of lipids in gases	Use of expensive process and equipment
Solvent Emulsification–Diffusion		
Solid lipid is dissolved in non-polar organic solvent and dispersed into the aqueous phase containing a surfactant forming an emulsion. The organic solvent is evaporated from the emulsion, under reduced pressure. As a result, the SLNPs are prepared in the aqueous phase [61]	Simple procedure Suitable for thermosensitive drugs	The insolubility of lipids in organic solvents. Thermodynamically unstable. Low lipid content, lack of scale-up
Double Emulsion		
The dissolved drug in the aqueous phase is added to the melted lipid and surfactant at a higher temperature. The microemulsion is then further added to a mixture of containing the water and surfactant in order to obtain a water/oil/water emulsion [62]	Suitable for hydrophilic and peptide-based drugs Surface modification of NPs by incorporating hydrophilic polymer	Multiple steps requirement Tends to form large particles.
Spray-drying		
In a one-step process, using an organic solvent, the lipid particles are dissolved, and the solution is then evaporated resulting in a dried particulate formation [41]	Cost-effective Single step process High size uniformity	Suitable for lipids with a melting point > 70 °C. Leads to particle aggregation formation as a result of high temperatures and shear forces.
Coacervation		
A mixture containing fatty acids salts and polymeric stabilizing agents is added in the aqueous phase which is then heated to a temperature to obtain a transparent alkaline micellar lipid salt solution. A suspension is obtained by gradually adding the coacervating solution into the mixture. The suspension is then cooled in a water bath under agitation resulting in the formation of LNPs. The drug is mainly dissolved in alcohol which is then added in the lipid phase or incorporated into the blank LNPs [63]	Solvent free method Easy to scale up	Not applicable for pH-sensitive drugs

## 3. Properties of Lipid NPs Impacting Performances

Understanding the key physicochemical properties of NPs is important for developing effective pharmaceutic products. Structural determinants such as the size of the particles and the presence of functional groups on the surface of particles are the critical elements which govern the delivery efficiency of these NPs.

### 3.1. Particle Size

The particle size of LNPs plays a vital role in their in vitro behaviour as well as their in vivo performance. Normally the average particle size of LNPs ranges between 100 and 400 nm and LNPs with particle sizes between 10 to 150 nm are applicable for systematic drug delivery via intravenous (IV) injection [64,65]. Generally, smaller NPs tend to aggregate often during dispersion, storage, and transport; however, due to their larger surface-to-volume ratio as a result of smaller size, they promote faster drug release. NPs with a size > 100 nm in diameter are easily taken up by the reticuloendothelial system (RES) in the lung, liver, spleen, and bone marrow, whereas smaller NPs face a prolonged circulation time in reaching to their target site [66].

### 3.2. Particle Surface Charge

The presence of surface charge on the LNP influences its interactions with cellular membranes, which is experimentally determined by zeta potential. The value of zeta-potential also indicates the colloidal dispersion stability by measuring the degree of repulsion force. Normally an LNP with zeta potential > +30 mV or <−30 mV is considered strong enough to repel and remain electrostatically stable and thereby helps in preventing the aggregation of LNPs [29,67].

Generally, the dimensions and charge density of the head group determine the surface charges on LNP. Depending on the type of lipids used, the surface charge could be anionic, cationic, or zwitterionic. The surface charge density governs the surface potential which controls the counterions adsorption and the nature of interactions between particles, hence determines the stability of NPs. Uncharged particles with low charge density often aggregate in a due course of time, while particles with higher charge density, strongly repel each other and thereby prevent aggregation [29]. The surface charge on LNP also helps to favour its interaction with the cell membrane and also facilitates the endosomal escape. The anionic cell membrane generally repels anionic LNPs, while the use of cationic LNPs causes cytotoxicity as it directly disrupts the cellular membrane. The use of ionizable lipids (ILs) in LNPs has been implemented since the overall surface charge on ILs is dependent on environmental pH, thereby helping to avoid any unwanted electrostatic interactions with the cell membrane.

### 3.3. PEGylation

PEG, a non-ionic biocompatible synthetic polymer, is soluble both in aqueous and non-aqueous solvents [68]. Incorporation of PEG provides the LNPs with an external polymeric layer onto their outer shell, and this helps to hinder the adsorption of serum proteins and the components of a phagocytic system, thereby extending their in vivo circulation time. It prevents aggregation during storage and increases the stability of the NPs. The length and density of the polymer chains determine the circulatory half-life of LNPs. The PEG chain length and its molecular weight (750 to 5000 kDa) has shown varying effect in vivo clearance study, which resulted from its interaction with different-sized opsonins which is present in the bloodstream [69].

The surface coating of LNP with PEG also influences the overall surface charge on LNPs. For instance, Kumar et al. studied the effect of polyethylene glycol-1,2-dimyristoyl-sn-glycerol-3-methoxypolyethylene glycol (PEG-DMG) for shielding the LNP surface charge using 1.5% (LNP1.5), 5% (LNP5), and 10% (LNP10) PEG-DMG concentration (Figure 3). The LNPs were formulated using 1,2-distearoyl-sn-glycero-3-phosphocholine (DSPC), cholesterol, DLin-MC3-DMA. The increasing PEG-DMG concentration resulted in a decrease in zeta potential (performed at pH 5.5) which was observed to drop from +32 mV (for LNP1.5) to +24 (for LNP5) and +18 mV (for LNP10) suggesting that higher PEG-DMG reduces the surface charge density of LNP. Furthermore, the combined analysis of particle surface charge with pH titration together with haemolytic activity suggested that the high PEG density provided a more significant physical steric barrier in the inhibition of membrane fusion and disruption, as compared to the charge shielding effect property.

The length of the acyl chain of PEG also regulates the intracellular delivery of a drug into the target cell. Usually, to the membrane of LNPs, the PEG-lipid is anchored using the hydrophobic acyl chain [62]. PEG-lipids having short acyl chains dissociate quickly from the LNPs following injection (since less energy is required in order to break the anchoring bonds present between the PEG-lipid and LNP), allowing the LNPs to better interact with target cells. For instance, PEG-lipids with C14 acyl chains take a half-time of around 1 h to dissociate from LNPs, while PEG-lipids with C20 acyl chains take 24 h or longer half-times for dissociation. PEG shields the LNP from the serum protein such as apolipoproteins (ApoE) and albumins. Judge et al. revealed that LNPs containing high amounts of C18 PEG-lipids, showed longer circulation time and thereby enhanced the efficacy LNPs as compared to LNPs with lower amounts of C14 PEG-lipids. The reduced association of ApoE was observed due to the highly shielded LNPs provided by mPEG200. [71]. This resulted in longer circulation which led to a greater opportunity of LNPs to enter subcutaneous tumour cells [72]. However, an excessive amount of PEGylation in LNP could inhibit cellular internalization and intracellular release of the drug, resulting in reduced intracellular delivery [73].

Suzuki et al. also studied the PEG shedding profiles of DMG-LNP and DSG-LNP, respectively for encapsulation of siRNA specific for GFP or FVII. DMG-PEG with shorter acyl chains displayed faster shedding from the LNPs than the DSG-PEG having the longer acyl chain. In addition, the PEG shedding rate also influences the production of anti-PEG antibodies and could pose complications in repeated administration. When different doses (0.003 to 0.3 mg/kg body weight) of DMG-LNP or DSG-LNP encapsulating siGFP were injected into mice intravenously. It was found that the production of anti-PEG IgM levels was significantly increased by DMG-LNP at a dose of 0.003 and 0.003 mg/kg body weight significantly. On the other hand, all doses of DSG-LNP led to the significant production of anti-PEG IgM. Therefore, it suggested that the anti-PEG IgM production is dependent on faster PEG shedding from the LNPs [74].

### 3.4. Surface Modification with Targeting Ligands

Another key strategy to enhance the delivery efficacy is the surface modification of LNPs by conjugating a targeting ligand. For this purpose, ligands such as antibody fragments, antibodies, and peptides specific to cell surface receptors are properly positioned at the periphery of the NP making it applicable for tumour-targeting. The most commonly used approaches for targeting include folate receptor (FR), α_V_β_3/5_ integrin receptors, α-transferrin receptor, and prostate-specific membrane antigen (PSMA). In many cancer cells, folate and transferrin are the most commonly overexpressed receptors, and therefore attachment of the corresponding ligands on the surface of LNPs has been extensively used for targeting cells or tissues. In addition, folate receptors are also overly expressed on the surface of macrophages which are involved in a no. of inflammatory diseases such as Crohn’s disease, psoriasis, rheumatoid arthritis, and atherosclerosis. In rapidly proliferating cancer cells, transferrin receptors are overexpressed to meet the increasing demand of iron, making it possible to develop transferrin receptor-targeted anticancer therapies.

Xue and Wong prepared folate-tagged LNPs to target folate receptors expressed on prostate cancer cells. It was found that folate-siRNA-LPN improved cellular uptake of siRNA by cells expressing the folate-receptor cells as compared with noncancerous RWPE1 cells and reduced the non-specific toxicity. Folate-LPNs not only promoted the extended-release of siRNA intracellularly but also extended in vivo RNAi activity [75]. Wang et al. prepared vitamin-derived LNPs (VcLNPs) with Vc lipid/1,2-dioleoyl-sn-glycero-3-phosphoethanolamine (DOPE)/cholesterol at a molar ratio of 30/30/40. VcLNPs was used for the encapsulation of mRNA, encoding a hybrid protein of antimicrobial peptides and cathepsin B. The mRNA delivery using VcLipid was ~70- and ~300-folds more efficient in comparison to Lipofectamine 3000 and electroporation, respectively. The schematic design of LNP is presented in Figure 4.

Wang et al. used NP3.47, as an inhibitor of the Niemann–Pick type C-1 protein (NPC-1), to conjugate the LNPs surface. The accumulation of siRNA with NP3.47 decorated LMPs (vs. unmodified LNP) was ~3-folds higher in late endosomes/lysosomes, indicating improved siRNA delivery [76]. The modified LNPs resulted in the efficient delivery of siRNA-EpCAM when targeted against epithelial cell adhesion molecule (EpCAM) Epi-1, positive cells in vitro and also minimized the progression of tumour in mice [77]. PEGylation can also serve for the functionalization of LNP bioconjugation with biomolecules or ligands. For example, Chen et al. reported that GalNAc conjugated with LNP-siRNA-PEG formulation, a single dose of administration displayed a significant gene silencing effect on hepatocytes Factor VII (FVII). These findings could be useful for treating hepatocellular carcinoma [78]. Similarly, Singh et al. used hyaluronan-conjugated DSPE-PEG-amine using carbodiimide chemistry for tumour targeting [79], and Parhiz et al. used DSPE-PEG-maleimide for conjugating antibodies using thioester linkages [80].

Beyond the usual components of LNPs (ILs, cholesterol, amphipathic phospholipids, and PEG-lipids), the addition of supplemental components (known as a SORT molecule) could also alter the in vivo delivery of cargo thereby helps in the improvement of tissue-specific delivery. Cheng et al. studied the effect of SORT NPs for tissue-specific mRNA delivery, wherein different classes of LNPs were engineered systematically via the addition of a supplemental SORT molecule to exclusively edit extrahepatic tissues. The four mDLNP formulations consisted of 5A2-SC8 (degradable dendritic cationic IL), cholesterol, DOPE, DMG-PEG (15/15/30/3 molar ratio), and mRNA. In this system, varying concentrations of permanent cationic lipid 1,2-dioleoyl-3-trimethylammonium-propane (DOTAP) were added. It was observed that the varying concentration of DOTAP was a key factor in determining tissue specificity, wherein 0% of DOTAP showed liver delivery, 10–15% showed spleen specific delivery, while 50% was optimal for lung-specific delivery. Similarly, the incorporation of 10–40% anionic lipids 1,2-dioleoyl-sn-glycero-3-phosphate (18PA) as a SORT molecule showed spleen selective delivery [81].

## 4. Biological Barriers to Lipid Nanoparticle Delivery

To accomplish their mission, LNP formulations of nucleic acids must overcome several biological obstacles. Nucleic acids need to be first protected against digestion in physiological fluids by nucleases, which occur through full encapsulation of nucleic acids by LNPs creating a physical barrier against nucleases. Secondly, designed LNPs should be able to evade the mononuclear phagocytic system (MPS) and renal glomerular filtration following systematic administration. Due to its intrinsic role in monitoring the body, MPS in the spleen and liver is a frequent destination for LNPs [82]. Lowering the clearance of NPs by MPS can prolong their circulation lifetime [83] most commonly accomplished by utilizing biodegradable groups to anchor PEG groups on NP surface [84], which reduces the opsonization by serum proteins and reticuloendothelial clearance [85]. Additionally, due to the increased permeability and retention (EPR) effect, PEGylated NPs are likely to extravasate from tumour vasculature to solid tumours [86]. PEG surface coating is exploited to govern the cellular uptake kinetics and prevent PEG-specific antibody induction by dissociating eventually. Dissociation of PEG is essential to prevent rapid systematic clearance of subsequent doses via accelerated blood clearance [87]. Adjusting the PEG structure can subside the accelerated blood clearance by attuning the kinetics of shedding and chain recognition [74,87]. The intracellular transposition of NPs, which is a crucial step for the transportation of siRNA in the tumour cells, is favoured by dePEGylation, on the other hand [83]. Ester motifs are one more strategy for enhancing biodegradability [88]. In addition to being chemically stable, easily incorporated, and bio-cleavable, ester moieties also have controlled biodegradation [89,90,91]. Upon surviving the filtration systems, LNPs should be able to reach their target cells subsequently and escape endosomal maturation upon internalization, which is believed to be facilitated by LNP’s hexagonal phase structure and pH-ionizable moieties [92]. LNPs fuse electrostatically to the cell membrane and use an inverted non-bilayer lipid phase to enter the cells by endocytosis [93]. The mechanism of cellular entry is governed by the physiochemical properties of nanocarriers which includes shape, size, surface charge, and surface composition [94,95]. Interestingly, LNPs can also be exocytosed (e.g., ~70% of LNP-siRNA formulations) [96] which gives rise to another challenge to LNP delivery [96]. Once present inside the cell, nanocarriers will be routed into early endosomes and then to lysosomes, the site of major metabolism, through maturation to multi-vesicular late endosomes. Otherwise, the internalized nanocarriers can be degraded during endo-lysosomal trafficking via recycling pathways [96,97]. The proton sponge hypothesis states that endosomal escape occurs as a result of gradual ATP-driven acidification from 6.5 to 5–6, promoting protonation of amine residuals in LNPs formulations, which allows cargo release following disruption of endosomal membrane [98]. The actual endosomal escape mechanism can be more complex and is influenced by a number of different variables, including endosome size, late endosome formation, membrane leakiness, localization of Rab7a on the endosomes surface, and activation of mTORC1 for downstream signalling for synthesis of proteins [99]. Finally, the nucleic acids, should be released (freed from the carrier) either into the cytoplasm (in the case of mRNA and siRNA) wherein the endogenous RNA interference machinery and translation of protein is located, resulting in up-regulation (mRNA) or down-regulation (siRNA) of target proteins, or should be released into the nucleus (in the case of pDNA) where transcription takes place [100,101,102].

The initial investigations using LNP-plasmid delivery showed limited success since this approach was impeded due to the inevitable requirement for nucleus entry. LNP technology, however, flourished in the case of siRNA delivery since this nucleic acid can function adequately if only recognized by RISC, which is located in the cytosol. This quicker approach led to the first vigorous gene silencing activity in nonhuman primates (NHPs) using stable nucleic-acid lipid particles (SNALPs) with siRNA payload, which were tailored against apolipoprotein B (ApoB) in 2006 [103]. Twelve years later, the first LNP-siRNA drug (Patisiran) was authorized by the FDA to treat hereditary transthyretin-mediated amyloidosis [104].

Multiple intracellular and extracellular barriers have been explained in the following section concisely, along with some related studies highlighting attempts to overcome the respective issues.

### 4.1. Liver Accumulation

One major feature, and limitation at times, with LNP delivery is their propensity for liver accumulation, where they are taken up by the RES [105]. Upon systematic administration [106], various serum electrolytes, proteins, and lipids will adsorb onto the LNP’s surface and form the so-called “biomolecular corona” [107]. This corona can significantly influence the journey of systematically administered NPs from biodistribution and cellular uptake [108] to systematic circulations and nano-bio interactions [109]. The composition of ionizable lipid particles can have a considerable effect on the formed corona [110]. ApoE is one of the most implicated serum proteins that play a significant role in the clearance and endogenous LNPs targeting to hepatic cells [72]. The delivery of nucleic acids to the liver is partly attributed to the organ’s well-perfused nature and its fenestrations as well [100]. It has been recognized that LNPs accumulate in different cells present within the liver [111,112,113]. Particle size, lipase sensitivity, and apparent pKa are some characteristics that govern the intrahepatic distribution of LNPs [114]. Chen et al. investigated the effect of particle size on influencing the tissue penetration and potential of LNP formulations of siRNA. They injected the LNPs intravenously into mice and found that regardless of the size, the majority of LNPs were found 24 h after injection in the liver. Less than 1% of the LNP formulations were found in the kidney, pancreas, lung, heart and femur. However, ~10% of the 80 nm size LNPs was found in the spleen. Since the particle size did not seem to be heavily involved in liver accumulation, they further investigated other parameters that can reduce the potency of small LNPs for silencing. In the end, they concluded that there was a clear hierarchy of LNP formulated siRNAs’ capability for gene silencing (78 > 42 > 38 > 27 > 117 nm). The LNPs with ~80 nm size demonstrated the maximum silencing activity, which was attributed to lower activity and stability of smaller-sized particles along with (as well as) the inability of particles larger than 100 nm to access the hepatocytes [115]. In another study investigating the relationship between LNPs’ physiochemical properties and efficiency of siRNA delivery to liver cells, Sato et al. once again showed a size-dependent depletion of gene silencing for 172 and 433 nm particles. In contrast, particles with size of 76.5 and 117 nm consistently showed greater gene silencing activity in hepatocytes. Interestingly enough, for targeting liver sinusoidal endothelial cells (LSECs), they concluded that by adjusting the LNP size around 200 nm, which is larger than the fenestrae size in mice, will lead to an increased specificity [114].

Regarding the pKa of ionizable moieties of LNPs, the same study demonstrated that the intrahepatic distribution of siRNA will be significantly changed due to small changes in pKa value, which will subsequently affect gene-silencing activity in hepatocytes as well as in LSECs. The authors further showed that the ED_50_ for gene silencing (i.e., effective dose for 50% silencing) vs. pKa curve in hepatocytes was bell-shaped, with maximum activity at a pKa of 6.45. At the same time, the same pKa value was not optimal in terms of specificity for hepatocyte gene silencing. They then concluded that since formulation with low pKa results in poor endosomal escape because of the inability to convert to cationic moieties in endosomes, a new mechanism must be established to determine the balance between activity and specificity in hepatocytes. In LSECs, on the other hand, a sigmoid curve was observed for gene expression versus pKa value, indicating improved gene-silencing efficiency with respect to rising pKa value [114].

Intrahepatic localization and activity of LNPs can also be modulated by lipid sensitivity to phospholipase. Based on the knowledge of the existence of extracellular lipases on the surface of liver cells’ membrane, including lipoprotein lipase (LPL), hepatic lipase (HL), and endothelial lipase (EL) [116], and the fact that HL is expressed only on the hepatocytes surface, and LPL and EL on the surface of LSECs [117], Sato et al. hypothesized that EL-sensitive cationic IL in the hepatocyte-specific LNP-siRNA systems is degraded by either LPL or EL present on the surface of LSECs but not on the surface of the hepatocytes (Figure 5). To reveal this involvement, they used GSK264220A (an inhibitor of both LPL and EL) and orlistat (an inhibitor of LPL only) as co-treatments. Considering that EL is principally an A1 phospholipase (PLA1) and GSK264220A inhibits EL activity [98], and that PLA1 activity of HL was much lower than that of EL [118]. It was suggested that the LNPs are being deactivated by the PLA1 activity of the EL [97].

A potential strategy for non-hepatocyte delivery is to deviate from ApoE-dependent pathways of delivery by increasing the PEG-lipid content in LNPs, which was not successful in terms of prolonged circulation and redirection to extrahepatic targets [119]. Conjugation of targeting ligands on the LNPs surface proved to be effective in facilitating the uptake by specific organs in small-scale settings. For instance, conjugated antibodies targeted against vascular cell adhesion molecules VCAM-1 or PECAM-1 and CD-4 were employed to redistribute the LNPs from liver to lung, cerebral endothelium during brain oedema and in all T cells (naïve, memory, central, and effector) in both lymph and spleen, respectively [120]. Moreover, successful localization of LNP-siRNAs to the antigen-presenting cells’ cytoplasm has been reported [121]. Another approach examined by Saunders et al. involved the pre-treatment of mice with a liposome that occupies liver cells temporarily prior to LNP delivery; this approach reduced the uptake of tested LNP-RNA formulations by the RES and ultimately led to enhanced bioavailability of the bioactive RNA, increasing the protein production for mRNA and better silencing for siRNA [105].

### 4.2. Spleen Accumulation

The accumulation of LNPs has also been noticed in the spleen upon systematic administration [111]. This has been attributed to protein adsorption on the surface of LNPs followed by surface opsonization and consequent uptake by splenic macrophages of the mononuclear phagocytic system [122]. Although targeting the spleen for LNP delivery can be considered as a promising approach for vaccine development, and oncology purposes [123], lipid and nucleic acid accumulation can trigger undesired immunological responses such as cytokine release syndrome in the spleen due to extensive production of IL-6 [124].

### 4.3. Maintaining Prolonged Protein Expression

Many studies on therapies related to enzyme and protein replacement, as well as interventions by siRNA-based therapies, have confirmed their potential for correcting genetic diseases. However, the problem with these approaches is their temporary nature [125]. Viral vectors and mRNA-encapsulating LNPs can be employed for long-lasting treatments to edit the genes by modifying the DNA itself through either loss or gain-of-function-related mutations [110]. Although successful in terms of its mission, viral vectors have not gained much interest for several reasons, such as excess cytotoxicity and immune reaction, the potential of off-target genomic integration, and its inability to administer repeated doses because of the host’s adaptive immunity towards the carrier. The LNP approach with mRNA as a nonviral vector, on the other hand, can produce permanent outcomes. Conway et al. managed to knock down the TTR or PCSK9 gene by over 90% by utilizing LNPs containing Zinc finger nuclease (ZFN) coding mRNA [126]. Another approach was the codelivery of mRNA along with a guide RNA inside an LNP in CRISPR-based studies, which showed promising outcomes in vitro and in vivo [127]. One example was recently demonstrated by Da Silva Sanchez et al. for cystic fibrosis treatment [128].

### 4.4. Immunological Responses

Nucleic acids can be recognized as invading pathogens in a host via various cellular sensors [129]. Synthetic siRNA can stimulate innate immune responses, especially in the presence of lipidic or polycationic carriers, which utilize endosomes to facilitate intracellular delivery [130]. Synthetic siRNAs have been found to be a potent inducer of inflammatory cytokines and interferons through Toll-like receptors [131] when used in nonviral delivery vehicles [132]. The immunostimulatory potential (potency) of nucleic acids is sequence-dependent, suggesting that the motifs can be modified for minimal immunostimulatory activity [114]. Chemical modifications of the nucleic acid can prevent the recognition of lipid-encapsulated siRNAs by pattern recognition receptors (PRRs) [130]. After the activation of innate immune responses, dsRNA-dependent protein kinase phosphorylates eIF2a, which downregulates the mRNA translation [133]. Modifying mRNA by N1-methyl pseudouridine increased both translational capacity and overall biological stability when evaluated in in vitro and in vivo mammalian cells. It also decreased its immunogenicity. It has been hypothesized that the synthesis of protein can be prohibited by RNA-dependent protein kinase which is activated by structural motifs present in the uridines base of mRNA [134], not in Ψ-modified mRNA. Superior translation might be also the reason behind enhanced stability via protecting the mRNA by high ribosome occupancy [135].

### 4.5. Endosomal Escape

Despite being recognized for a long time, endosomal escape remains one of the unresolved bottlenecks in the way of effective LNP design [136]. Following the cellular entry, LNPs will be trapped in endosomes, from which only a small fraction may be able to successfully escape. It has been estimated that only 2% of designed RNA delivery systems can escape the endosomes effectively [137]. Endosomal escape can occur via membrane fusion, rupture, or pore formation [98]. Numerous new formulations of LNPs are continuing to be reported that can more efficiently overcome endosomal entrapment. To better understand the endosomal escape steps, Herrera et al. employed a screening method which is based on a Gal8-GFP reporter fusion (Gal8-GFP) [138] to create a strong galectin 8-GFP (Gal8-GFP) cell reporter platform. This was used to directly visualize the potentiality of LNP-encapsulated mRNA for endosomal escape. As an indicator of the cytosolic availability of mRNA, this sensor system uses rapid and sensitive differences in endosomal membrane integrity [139]. Modelling of the delivery process was recently tackled by Mihaila et al. who designed an ordinary differential equation-based model which was used as a predictive tool for the optimization of the LNP-mediated delivery of siRNAs. This mathematical model can be used as an effective screening tool to compare the relative kinetics of different types of LNPs towards choosing the most efficacious choice prior to hands-on experiments. This model employs many critical steps of the intracellular RNAi pathway involved in the delivery (i.e., cell entry through the plasma membrane, endosomal escape and unpackaging, siRNA loading onto RISC, and mRNA breakdown) to predict the knockdown efficiency induced by novel LNP formulations of siRNA in vitro [28].

The process of endosomal escape has not been completely understood, but it is clear that cationic lipids may promote the fusion by increasing the electrostatic interactions with anionic endosomal membrane components leading to the cargo leak to the cytoplasm [92]. ILs are unique as at physiological pH they acquire a neutral charge while being protonated at low pH and thus becoming cationic, which can promote endosomal membrane destabilization and facilitate endosomal escape [140]. Numerous cationic ILs with various structures have been and are being developed, which share certain core characteristics:(i)Headgroups containing tertiary amines which are uncharged (zwitterionic) under physiological pH and become protonated at acidic pH [92],(ii)Lipid tails that promote self-assembly into a nanoparticle due to hydrophobic association. The tail properties can further affect the endosomal escape capability of LNPs. For instance, due to the stronger protonation at endosomal pH, branched-tail lipids demonstrate improved endosomal escape in comparison with their linear counterparts [88]. Lipid type and ratio can also enhance endosomal escape [141,142,143,144,145,146].(iii)Protonated lipids which contribute to an elevated propensity for membrane fusion in acidified endosomes in target cells [92]. Optimizing the pKa values of the ILs can positively affect the endosomal escape. Alabi et al. showed that among the three key variables, LNP size, LNP pKa, and siRNA entrapment, the strongest correlation with overcoming the biological barriers and consequently gene silencing capability was related to the pKa. They demonstrated that LNPs with pKa lower than 5.5 were not successful in gene knockout in vitro and in vivo systems [147].

The idea that conjugation of IL can serve as a strong determinant for siRNA pharmacokinetics was first put forth by Biscans et al. They covalently attached the IL, DLin-MC3-DMA, with siRNA and reported increased endosomal escape (evidenced by a 51% increase in large foci Gal8+ cells) in cell culture without disturbing the siRNA efficacy. They, however, observed a non-specific regulation of gene expression in tissues with more than 20 pmol/mg accumulation of DLin-MC3-DMA suggesting the limitations of this approach [148]. To overcome the limitations of ILs (as well as lipidic and polymeric systems in general), such as cytotoxicity and potential immune activation, hybrid delivery systems have been recently introduced [13]. Sanghani et al. introduced pH-sensitive PEGylated CL4H6-MRTF-B siRNA-loaded LNPs to safely deliver myocardin-related transcription factor B (MRTF-B) siRNA and efficiently into human conjunctival fibroblasts thereby preventing the conjunctival fibrosis after glaucoma filtration surgery. Their near-neutral PEGylated nanoparticles were non-toxic at 50 nM siRNA concentration while having a far superior silencing effect in comparison to their highly cationic non-PEGylated counterparts (>80% in vitro), which was attributed to effective endosomal escape [149].

By systematic derivatization of one of the previously developed lipids (YSK12-C4), structure-activity relationships (SAR) assisted in identifying the best pH-sensitive cationic lipid for further investigations. The SAR studies revealed that the apparent pKa is considerably affected by the hydrophilic headgroup structure, but not the hydrophobic tail. Thus, the endosomal escape (pKa dependent) and intrahepatic distribution (pKa independent) can be improved by the modifications of the headgroup and the tail, respectively. Notably, the hydrophilic headgroup and the hydrophobic tail minimally interact with one another, allowing for the independent use of both structures to design the desired pH-sensitive cationic lipid. They then formulated an LNP with a potent pH-sensitive cationic lipid CL4H6 (CL4H6-LNPs), which in in vivo experiments demonstrated more ability for endosomal escape, cytosolic release, and RNA-induced silencing when compared to the previously reported LNPs. The new formulation was also superior in terms of biodegradability and compatibility [150].

LNPs’ surface charge is an additional feature that can be tailored for targeting abilities. Anionic carriers have been utilized for brain therapeutics; in a comparative study, Gabal et al. reported 1.2-fold higher brain targeting efficiency for anionic nanostructured lipid carriers as compared to their cationic counterparts [151]. However, anionic particles experience limitations due to difficulties faced in nucleic acid packaging and poor transfection efficiency. Tagalakis et al. showed that PEGylation enhances the receptor-mediated transfection efficiency of anionic nano complexes. They used cationic targeting peptides as a connecting bridge between pDNA and PEGylated anionic liposomes. Not surprisingly, the newly produced structures displayed more resistance to aggregation in both serum as well as in transfected cells. They also demonstrated increased tissue penetration and broader cellular transfection than homologous non-PEGylated anionic and cationic systems [152]. Anionic integrin-targeted hybrid nanocarriers were also explored for siRNA treatment of neuroblastoma with reduced systemic and cellular toxicity and minimal clearance by the liver. Anionic receptor-targeted nano complexes were as specific and efficient as their cationic equivalent. This was evident in an animal model as well since anionic receptor-targeted nano complexes transfected tumours in an integrin-mediated fashion and entered tumours effectively, with little off-target biodistribution [153].

### 4.6. Cytotoxicity

A constant theme in the development of LNPs is the incorporation of lipid degradability to improve biocompatibility. Maier et al. sought to further optimize the LNP platform in this regard (e.g., higher capacity to be metabolized, increased in vivo transfection efficacy, and no toxic metabolites/by-products) by probing novel ILs used in LNPs. Following a review of the literature, they established a correlation between certain structural parameters and activity, which served as a roadmap for the development of effective lipids in vivo. Their lipid was amphipathic and contained a hydrophilic headgroup which is made up of an ionizable amine with long hydrophobic dialkyl chains. It also had a pKa between 6.2 and 6.5 and the ability to acquire a “cone”-like shape in the acidic environment [154,155,156]. They designed the lipid structure so that bio-cleavable groups were located within the hydrophobic lipid tails to introduce biodegradable functionality in such a way that would rapidly promote the in vivo metabolism into highly water-soluble products while maintaining excellent efficacy. They used an ester of linoleyl chain (L319) to replace the 9,10-cis double bond in order to keep the structural characteristics of the lipid necessary for in vivo activity. They reported considerable tolerability, facilitated excretion and elimination, and excellent potency in rodents and nonhuman primates (NHPs) [89]. Ester linkages were incorporated in LNPs’ tails, between C9 and C10 in the linoleyl chain (named L319), as they were easier to hydrolyze by intracellular lipases or esterases. This modification led to results that were on par with those of the highly potent MC3-lipid but with almost total elimination in a 24 period. The position of the ester bond played a vital role in the functioning and elimination rate: the closer the ester bond was to the head group, the greater its effect would be on the apparent lipid pKa, and the lower its silencing potency in vivo. The further the ester bond was from the headgroup, the more persistent the lipids would be in the liver [135].

Another approach to reduce the persistence of LNPs was the employment of disulphide bonds. Shirazi et al. synthesized a series of degradable multivalent cationic lipids (CMVLn, *n* = 2 to 5). Between the headgroup and the hydrophobic tail, a disulphide bond spacer was also introduced. This spacer can be easily cleaved upon the addition of reducing agents such as glutathione in the reducing milieu of the cytoplasm and thus facilitate elimination. The lipids transfected the mouse fibroblasts with comparable efficacy to highly effective non-degradable analogues and standard commercial reagents such as Lipofectamine 2000 while being much less cytotoxic [157]. Another derivative was developed by Akita et al. who designed a hepatocyte-targeting siRNA carrier by incorporating tertiary amines and disulphide bonds (ssPalmE) in the hydrophobic scaffold. They reported improved gene knockdown in the presence of vitamin E in the scaffold [158].

Finally, it must be noted that pH-sensitive ILs used in common LNP formulations are also advantageous for nucleic acid delivery from the cytotoxicity perspective, given their lower interactions with blood cells’ anionic membrane in a neutral state (at the pH of the circulation system) and thus higher biocompatibility [140].

### 4.7. Post-Administration Reactions

One other roadblock in the LNP-mediated delivery of nucleic acids is the undesirable post-administration reactions. The intravenous injection of LNPs can trigger both complement-dependent and complement-independent [159,160] reactions such as symptoms of mild flu or severe cardiac anaphylaxis [159]. Hypersensitivity reactions may be influenced by numerous physiochemical properties such as lamellarity, surface charge, and cholesterol content of LNPs [161]. A combination of corticosteroid immunosuppressant dexamethasone, for instance, antihistamines such as H1/H2 blockers, and oral acetaminophen, along with reduced infusion rate can be used as a pre-dosing action to manage the infusion-related reactions [162]. It has also been reported, in the case of Onpattro^®^, that the severity of the symptoms will subside by repeated administration and exposure to the drug [163]. Another approach is to incorporate PFG-lipids to decrease the possibility of LNPs’ recognition by MPS and renal filtration clearance [164].

## 5. Therapeutic Applications of LNP Formulations of siRNA

### 5.1. Acute Myeloid Leukaemia (AML)

AML is deadly cancer among children and elder adults. It is known that long noncoding RNAs (lncRNAs) play a vital role in regulating the cell cycle, which acts either directly as an oncogene or indirectly interacts with other oncogenes. As such, lncNAs can be used as therapeutic agents. Transcriptomic analysis of children having AML identified lncRNAs (LINC01257) which especially harbours the t(8;21) chromosomal translocation (leukaemia subtype termed as AML1-ETO) which is highly expressed in AML. Cornnerty et al. prepared cationic/stealth phospholipid LNPs (LNP-si-LINC01257) for delivery for siRNA (Figure 6). The delivery was safe with a robust LINC01257 silencing effect that led to an efficient reduction in AML cell proliferation. No toxicity towards healthy blood cells was reported. These findings suggested the use of the LNP-assisted RNAi mechanism for targeting cancer-specific lncRNAs as a feasible therapeutic strategy [165].

The chimeric fusion oncogene termed as *BCR-ABL* is a type of leukaemia-specific fusion transcript that mainly appears in human chronic myeloid leukaemia (CML) patients. Targeting siRNA against *BCR-ABL* oncogene could specifically knock down the fusion gene and thereby control cell proliferation. Jyotsana and co-workers formulated DLin-MC3-DMA LNPs for siRNA delivery targeting the *BCR-ABL* oncogene. It was observed that the formulated LNP efficiently delivered the siRNA both in vitro and in vivo, displaying nearly 100% uptake of LNP-siRNAs was observed in the bone marrow of a leukaemia model without any toxicity [166].

### 5.2. Breast Cancer

Insulin-like Growth Factor 1 (IGF-1) plays an essential role in the development of many tissues. Additionally, the IGF-1 receptor (IGF-1R) signalling pathway has been shown to play important role in numerous human cancers, including breast cancer [167,168]. Therefore, blocking the IGF-1R signalling could reduce cell proliferation and induce programmed cell death. Studies showed that using antioxidants such as Lycopene has the potential of preventing cancer in cell culture, in animal studies, and also in clinical trials [169]. Mennati et al. formulated Lycopene along with an anti-IGF-1 siRNA in mPEG-PCL-DDAB NPs to suppress IGF-1R in MCF-7 breast cancer cells. The combination of treatment in LNPs significantly reduced the expression level of IGF-1R (more so than the individual agents) leading to increased apoptosis induction in the MCF-7 cells. The result suggested that the co-delivery of lycopene with siRNA-loaded mPEGPCL-DDAB NPs can be considered a functional treatment for breast cancer [170]. In another study, different types of siRNAs (HSP90, CDC20, Mcl-1 and Survivin) were explored against MDA-MB-436 breast cancer cells [171]. The combination of Mcl-1 and surviving siRNA led to the suppression in the growth of cancer cells more effectively in comparison to single siRNAs [171,172].

Triple-negative breast cancer (TNBC) is known to be refractory to major drugs used in the treatment of breast cancers because this sub-type of breast cancer does not express the usual drug-targeted receptors such as progesterone, oestrogen, or human epidermal growth factor receptor 2 (HER2). Okamoto and co-workers developed an LNP which was modified with a Fab′ antibody for targeting heparin-binding epidermal-like growth (αHB-EGF) and used for the encapsulation of siRNA (αHB-EGF LNP-siRNA) for TNBC treatment. αHB-EGF binds to EGF receptor (EGFR) which is highly expressed in various cancer cells. αHB-EGF LNP-siRNA targeting the polo-like kinase 1 (PLK1) was formulated in order to evaluate its effectiveness on MDA-MB-231 TNBC cells overexpressing the HB-EGF. After injection, the expression of PLK1 protein was almost reduced in the tumour cells, making αHB-EGF LNP as a promising carrier and could be used for the treatment of the HB-EGF-expressing TNBCs as shown in Figure 7 [173].

### 5.3. Liver Disease

Non-alcoholic steatohepatitis (NASH) is a chronic liver disease which is known to be caused as a result of obesity. The nuclear High Mobility Group Box 1 (HMGB1) protein is secreted by damaged macrophages, hepatocytes, dendritic cells, monocytes, and other cells following liver damage and plays a vital role in the development of inflammation in NASH. Zhou et al. constructed a Mannose-modified HMGB1-specific siRNA (mLNP-siHMGB1) in LNP in order to target liver macrophages using mannose receptor intervention. The LNP was composed of DSPC, DLinMC3-DMA, DSPE-PEG-Man, PEG-DMG, and Cholesterol. The mLNP-siHMGB1 targeted via mannose receptor silenced the HMGB1 effectively and resulted in a reduction in HMGB1 protein in the liver. The combined treatment with mLNP-siHMGB1 and docosahexaenoic acid after 8 weeks, the liver functions of NASH mice was restored to its normal levels, making the combination promising for the clinical management of NASH [174] (Figure 8).

Sato et al. formulated LNPs using pH-sensitive cationic lipids YSK13-C3/chol/DMG-mPEG2k for siRNA delivery and modified them with N-acetyl-D-galactosamine (GalNAc), which is a hepatocyte-specific ligand. The attachment of GalNAc ligand resulted in increased specificity towards hepatocytes with a reduction in toxicity. The PEGylated GalNA-LNPs reduced the LNP-associated toxicity without affecting the gene-silencing activity of siRNA. A single injection of LNPs significantly reduced the hepatitis B virus (HBV) genomic DNA and antigen in chimeric mice with humanized livers [114].

ApoB is known as an essential protein expressed mainly in the liver and jejunum and is required for the transportation and metabolism of cholesterol. Zimmermann et al. encapsulated siRNAs (*APOB*-specific siRNAs) for targeting apolipoprotein B (ApoB) using non-human primates. The siRNA was encapsulated in liposomal formulation 3-N-[(q-methoxypoly(ethylene glycol)2000)carbamoyl]-1,2-dimyristyloxy-propylamine (PEG-C-DMA), DSPC, DLinDMA, and cholesterol (2:40:10:48 molar ratios) for silencing the ApoB in non-human primates (cynomolgus monkeys). A single IV administration resulted in the silencing of ApoB mRNA in a dose-dependent after 48 h of administration in the liver, with a >90% silencing effect. Additionally, the reductions in ApoB protein, low-density lipoprotein, and serum cholesterol levels were also observed [103].

### 5.4. Hepatitis B

Hepatitis B is a severe form of liver dysfunction which is caused by HBV. In 2019, Hepatitis B was estimated to cause 820,000 deaths, mostly due to cirrhosis and hepatocellular carcinoma (HCC). A novel IL, LC8 (five-tailed amine compound), was developed and used for investigating siRNA delivery. The LNPs formulation (RBP131) consisted of LC8, cholesterol and DPPE-mPEG2000 (16:0 PE2000 PE) (Figure 9A,B). The pKa value of RBP131 was found to be 6.21, fulfilling the required pKa (6.0–6.5) for delivery materials, helping the siRNA from endosomal space in a pH-dependent manner. The efficiency of RBP131 to deliver siRNAs targeting ApoB or HBV gene was evaluated. In RBP131, the siRNA encapsulation efficiency was very high (>80%). RBP131/siRNA NPs size varied between 60 to 100 nm with a neutral (0.089 mV) value of the zeta potential in the PBS buffer. The efficiency of RBP131 for delivering siRNA against ApoB and the gene silencing ability exhibited excellent siRNA delivery efficiency, with a median effective dose (ED_50_) of ~0.05 mg/kg employing the anti-ApoB siRNA [175].

Worldwide, HBV-induced HCC is the second most major cause of death related to cancer. It is known that serine/threonine polo-like-kinase 1 (PKL1) is a pivotal factor in HBV infection and is overexpressed in many human cancers. In 2020, Foca and group used siRNA for targeting the antiviral activity of PLK1. The LNPs were formulated using 1.2-dipalmitoyl-sn-glyceo-3-phosphocholine,1,2-dilinoleyloxy-N,N-dimethylpropylamine, 3-N-[ω-methoxypoly(ethyleneglycol)_2000_) carbamoyl]-1,2-dimyristyloxy-propylamine and cholesterol. The LNP-siPLK1 resulted in reducing the amount of secreted viral particles on HBV-infected primary human hepatocytes [176]. In the liver sinusoidal endothelial cells (LSECs), the accumulation of LNPs resulted in the secretion of several cytokines which are then followed by neutrophilic inflammation. To overcome this, Sato and groups prepared the LNPs which were modified on the surface with N-acetyl-D-galactosamine (GalNAc), a hepatocyte-specific ligand. The modified LNPs improved the hepatocyte specificity and together with PEGylation displayed a reduction in the LNP-associated toxicity without affecting the gene-silencing activity in hepatocytes. It was evident that a single injection of LNPs, significantly reduced the HBV genomic DNA and antigen in chimeric mice with humanized livers infected with HBV without displaying any sign of toxicity [177].

### 5.5. COVID-19

At the end of 2019, a new virus variant known as Severe Acute Respiratory Syndrome Coronavirus 2 (SARS-CoV-2) first originated in Wuhan city of China. As the virus was highly transmissible, the World Health Organization (WHO) quickly declared the SARS-CoV-2 disease (subsequently named COVID-19) as a global health emergency. This virus globally claimed the lives of more than 6 million people to date. In the development of a therapeutic agent, the main targets of SARS-CoV-2 include surface spike protein and/or RNA [178]. The inactivation of spike protein can be achieved by surface engineering of contacting surfaces [179] and physical and chemical sterilizations [180,181]. Researchers are also exploring new therapies using RNAi technology to target the RNA of SARS-CoV-2 [182,183]. As there are other reviews on LNPs for mRNA vaccinations against the COVID-19 disease [184,185,186], we are not covering this aspect of LNPs in this review.

Wu and Luo using Mfold web server, developed a method for designing the effective siRNAs against the three viral genes (spike, membrane, and nucleocapsid gene) using the concept of probability of binding efficiency (PBE) combined with the RNA secondary structures. using these 11 types of siRNAs were designed which then targeted against the consensus regions of the three key genes. The silencing efficiency of the designed siRNAs was tested by transfecting HUVECs and A549 cells with plasmid DNA expressing the S gene, which was subjected for 6 h using liposomal Lipofectamine 2000 as a carrier. The designed siRNAs (3329i, 1878i, 1104i, and 2351i) were found to efficiently reduce the mRNA levels of the target protein in both cell lines. In HUVECs cells, the 3329i siRNA (highest value of PBE) resulted in a 70% reduction in the mRNA expression while 2351i and 1878i siRNAs lowered the mRNA expression by 50%. In A549 cells, 3329i siRNA displayed a 55% reduction in S gene expression, with similar silencing effects displayed by the other three siRNAs. The results suggested the usefulness of the PBE value for designing siRNAs. For the N gene, the top four (418i, 881i, 214i, and 1068i) siRNAs were selected based on high PBE values and compared the silencing effects in two cell lines. In HUVEC cells, 418i siRNA (highest PBE value) reduced the N gene expression by 64%, while the remaining 3 siRNAs showed a 55–61% reduction in gene expression. In A549 cells, 418i and 1068i siRNA having high PBE values displayed a better silencing effect comparatively. Similarly, for M gene targeting, 607i, 344i, and 79i siRNAs were used. In both HUVECs and A549 cells, 607i siRNA lowered the expression of the M gene by 39–43%, while the other siRNAs lowered the expression of the M gene by 39–47% in A549 cells and by 26–30% in HUVECs cells. The overall results concluded that most of the designed siRNAs are capable of reducing viral gene expression, with >50% inhibition rates [187].

Idris et al. used DOTAP/MP3 LNP-siRNAs (dmLNP-siRNAs) for targeting SARS-CoV-2 (Figure 10). The lipid formulation consisted of DOTAP:MP3:DSPC:Cholesterol: C16-PEG in a 40:25:10:22:3 molar ratio. The average size of dmLNP was 80 nm with a zeta potential of ~18 mV. The siRNA used was siHel1, siHel2, UC7, and siUTR3 in K18-hACE2 mice (model mice). Firstly, the mice were inoculated with 1 × 10^4^ plaque-forming units (PFUs) of the virus followed by intravenous treatment with various sLNP-siRNA formulations a day before and two days after the inoculation. When compared to virus-infected mice and dmLNP-siRNA control-treated mice, the treatment of mice with dmLNP-siHel2 and dmLNPsiUC7 provided a survival advantage and exhibited less weight loss, accompanied by an overall decreased clinical score on 6 to 8 days, suggesting that the treatment with siRNAs may reduce the severe disease symptoms. Additionally, in the in vivo study, based on lung analysis of viral outgrowth, the treated mice with siRNAs were able to functionally repress the SARS-CoV-2 expression at 7–8 days. Thus, their overall studies suggested that dmLNPs bearing siUC7, siUTR3, and siHel2 siRNAs were capable of repressing SARS-CoV-2 expression in vivo, and also in delaying the onset of COVID-19 symptoms and therefore could as used as a potent siRNAs as a therapy [188].

Tolksdrof and co-workers, using Horizon siDesign tool, designed and tested 8 siRNAs for targeting the highly conserved 5-untranslated region (5-UTR) of SARS-CoV-2, each of which was capable of downregulating the viral gene activity. Hela cells were transiently co-transfected using the dual-luciferase reporter (psiCheck2 SARS-CoV-2 50 -UTR) along with one of the eight different siRNA candidates. It was observed that all designed siRNAs targeted the 5-UTR of SARS-CoV-2 efficiently and significantly downregulated the luciferase reporter activity. siCoV6 was the most efficient siRNA among the 8 siRNA as it was able to bind to the highly conserved transcription regulatory sequence (TRS) and thus targeted the leader sequence [189].

In silico approaches are being utilized for designing more efficient siRNA agents against SARS-CoV-2, Bappy et al. designed a siRNA to inhibit the N-gene of SARS-CoV-2 that is involved in viral replication, encapsidation, and RNA packaging. Using the SiDirect 2.0 server, the siRNA was designed following the prediction of Guanidine Cytosine (GC) content and secondary structures using the Mfold and OligoCalc servers. The verification of considered siRNA was performed using the siRNA Pred server. The hybridization energy and base-pairing pattern were calculated using the RNAcofold server provided by Vienna RNA web services for the selected RNA sequences (Table 3). The GC contents of the proposed siRNA were found to be within the desired range (30–52%) and also the binding free energy of the four siRNA was within the qualified range, thereby suggesting efficient RNA-RNA interactions. Sequences 1 and 3 had a score of more than >0.9 and displayed very high efficacy. Sequence 2 had a score between 0.8 and 0.9 and showed high efficacy, while sequence 4 scoring less than 0.8 indicated moderate efficacy. The designed siRNA molecules were proposed to be an alternative therapeutic approach against the various Bangladesh strains of SARS-CoV-2 [190].

Sawan et al. employed RdRP gene as the siRNA target, which codes for RNA-dependent RNA polymerase enzymes in SARS-CoV-2. The RdRp gene is a multi-subunit replication/transcription complex involved in viral replication. The siRNAs were designed using bioinformatics-based steps. A single effective siRNA molecule (guide: 5′-UAGUACUACAGAUAGAGACAC-3′; passenger: 5′-GUCUCUAUCUGUAGUACUAUG-3′) was selected and used for studying molecular docking and molecular dynamics simulation studies. The outcome suggested that the designed siRNA could be effective against SARS-CoV2 RdRp [191]. Choudhury et al. designed siRNA molecules against the surface glycoprotein genes and nucleocapsid phosphoprotein of SARS-CoV-2. The conserved sequences from 139 SARS-CoV-2 strains were collected globally and were utilized for the construction of specific siRNAs. A total of 34 conserved regions (15 nucleocapsid phosphoprotein and 19 surfaces glycoprotein) were identified. After analysing the free energy of folding and binding, GC content, melting temperature, efficacy prediction, and analysing molecular docking, eight siRNAs were selected for the best action. While promising computationally, further studies are required for therapeutic validation [192].

Chen et al. also performed theoretical predictions of potential siRNAs for targeting the SARS-CoV-2 genome. The representative SARS-CoV-2 genome (MN908947) and mutational landscape information were collected from NCBI and 2019nCoVR databases. Using a single reference genome which was obtained from NCBI and by analysing single nucleotide polymorphisms (SNP) at the target sites, nine siRNAs were identified for five different sites of the SARS-CoV-2 genome having 21–25 nts of length. Out of the nine target sequences, only two SNPs, i.e., target 1 (‘AAUAGUUUAAAAAUUACAGAAGA’) and target 2 (‘CAACUAUAAAUUAAACACAGA’) were found. For target 1, a single SNP was present in the BetaCoV/Wuhan/HBCDC-HB-05/2020 strains, and for target 2, the SNP were found to be present in BetaCoV/Singapore/2/2020 and BetaCoV/Singapore/6/2020 strains. The first target is a coding sequence variant while target 2 is a missense variant (G to A) resulting in a different sequence of amino acids. The results obtained suggested that both of the selected targets (1 and 2) possess the conserved sequences among the current SARS-CoV-2 genomes [193].

Patisiran (developed by Alnylam with a trading name Onpattro, Cambridge, MA, United state) was the first siRNA used for the treatment of hereditary transthyretin (hATTR) amyloidosis using LNP-based siRNA drug, approved by the U.S. Food and Drug Administration in 2018. Their siRNA is designed to target a sequence present within the untranslated region (3′ UTR) of the transthyretin (TTR) mRNA using LNPs as a delivery vehicle (Figure 11). The siRNA is modified by partially replacing uridine and cytosine with 2′O-methyl uridine and cytosine. The LNPs are formulated using DLin-MC3-DMA: DSPC: PEG_2000_-DMG: Cholesterol. The LNP-based siRNA drug was administered intravenously every three weeks at a dose of 0.3 mg/Kg, which accumulated in the liver as it was the primary site for the production of the transthyretin (TTR). After the absorption of LNPs by hepatocytes, siRNA proceeds to degrade TTR mRNA thereby decreases the TTR protein production. This leading ‘siRNA company’ Alnylam also developed a siRNA formulation for COVID-19 treatment using the same strategy [194].

## 6. Summary and Future Perspective

The LNPs represent an efficient and versatile platform as a delivery vehicle of nucleic acid therapeutics. Owing to their relatively biocompatible components and unique properties, LNPs have increased the therapeutic potential of siRNA in several indications. Their use in mRNA vaccines provided a very powerful thrust for the clinical translation of LNPs in other indications. It must be noted that vaccine formulations with LNPs bear a relatively low bar to satisfy since local deposition (rather than systemic administration) combined with non-specific uptake by immune cells (rather than cell-specific uptake) are sufficient for the required antibody generation. The deployment of RNAi with siRNA will require systemic administration and tissue-specificity for a successful outcome with robust silencing of a target gene in a specific cell population. Various types of LNPs such as liposomes, SLN, NLS, and NEs are available that, depending upon the applications, could be employed to formulate the siRNA. This review has highlighted the potential of siRNA through integration with surface-engineered LNPs for the treatment of cancer and other infectious diseases. This review has summarized the preparation methods and properties of LNPs along with the implications of different surface modification strategies. Surface engineering using targeting ligands such as antibodies, antibody fragments, and peptides imparts specificity and stability against biological barriers. LNPs as a delivery vehicle have successively delivered the therapeutic siRNA by stabilizing the nucleic acid, and therefore have widened the use of siRNA for numerous applications, thereby improving the treatment of cancer and other infectious diseases. The design of siRNA and its functionalization using surface-modified LNPs have been discussed for various applications such as acute myeloid leukaemia, breast cancer, liver disease, hepatitis B, and COVID-19. Furthermore, global efforts are exclusively devoted to the development of siRNA therapies against SARS-CoV-2 using a computational approach and have shown promising data which could help in developing an efficient treatment for COVID-19 in the near future.

An ideal LNP system must meet multiple requirements such as particle size in the order of 80 nm, sufficient stability and penetration through the liver fenestrae, and lack of inflammatory reactions, immunogenicity, and cytotoxic effects in a host. Other biological barriers include protection by nucleases, evading the MPS, and renal glomerular filtration for effective systemic distribution. The surface engineering of LNPs offers stability and target efficiency, which in turn improves the efficacy of loaded therapeutics, i.e., siRNA. The apparent pKa value of IL in the range of ~6.4 with a neutral surface charge is desirable to avoid sequestration by MPS. Thorough investigations of the ideal (or desirable) LNP structure(s), however, remain to be fully explored in different indications. It is possible that certain LNP configurations might be more advantageous for certain indications (e.g., provide better delivery at the disease site), but this issue remains to be investigated in a systematic way. The urgency to push for clinical use sometimes de-emphasizes such explorations, limiting clinically useful LNP formulations to a handful of formulations. With an increasing understanding of the contributing properties of the LNP components, further modification of the components could be employed in order to achieve a stable formulation and effective targeting of desired organs. The better implementation of organ-specific targeting is bound to enhance the efficacy of treatment as well as reduce any possible side effects by restricting the biodistribution of the drug and carrier to the site of action. This endeavour will be especially important to expand the utility of siRNAs beyond liver diseases (all siRNA approved to date are for liver indications).

For further improvements in bioavailability, increased circulation in the blood, delivery to the target organs, the surface modification of LNPs such as PEGylation, and incorporation of surface ligands on the LNPs have been found to be effective strategies. Engineered approaches to ‘shed’ the incorporated targeting/functional moieties will be beneficial to enhance the therapeutic effects once the carrier reaches its destination. Improved delivery in the order of 2- to 3-folds will not possibly be beneficial for improved clinical outcomes, but a more significant improvement such as 10-fold increased delivery might be required for significant clinical effects. It should also be noted that some of the effects from targeted systems may be derived from sites other than the intended target, so improved targeting to a particular site may reduce such ‘adjuvant’ effects of the therapeutic agent. As an example, the bone marrow deposition of LNPs used to treat subcutaneous solid tumours (as in most animal models) may induce immune stimulation that could contribute to the anti-tumour effects and restrict the biodistribution of LNPs to tumours, which may diminish the therapeutic effect from immune stimulation. One must be careful in engineering the LNPs for an effect in order not to lose other beneficial effects that are not readily obvious.

## Figures and Tables

**Figure 1 pharmaceutics-14-02520-f001:**
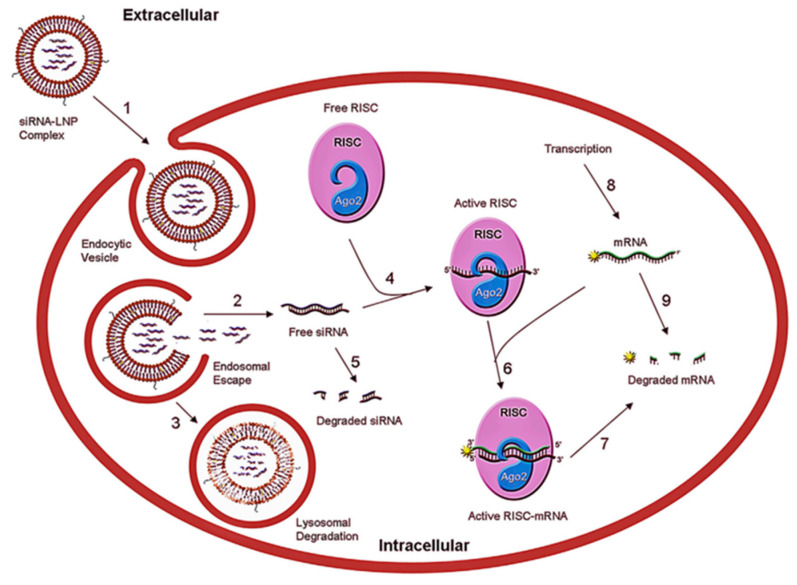
Schematic representation of mechanism of LNP-mediated siRNA delivery into cells resulting in gene silencing. The steps involve (1) LNP crossing the plasma membrane; (2) endosomal escape; (3) lysosomal degradation of LNP; (4) siRNA loading onto RISC; (5) siRNA degradation; (6) active RISC with target mRNA formation; (7) RISC cleaves the target mRNA; (8) mRNA transcription; and (9) mRNA degradation. Adopted from Ref. [28], Mol. Ther. Nucleic Acids 2017.

**Figure 2 pharmaceutics-14-02520-f002:**
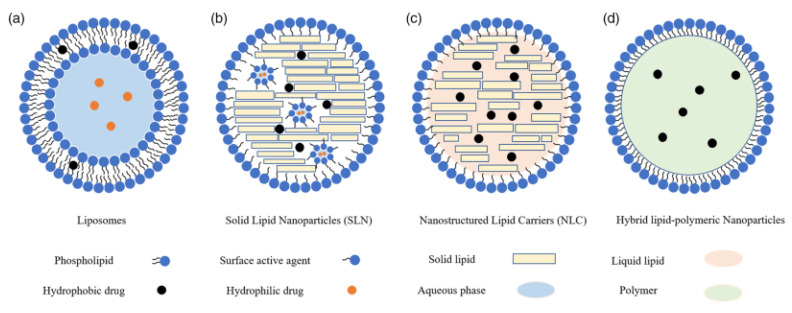
Different types of LNPs. Adopted from Ref. [30], Adv. NanoBiomed Res. 2022.

**Figure 3 pharmaceutics-14-02520-f003:**
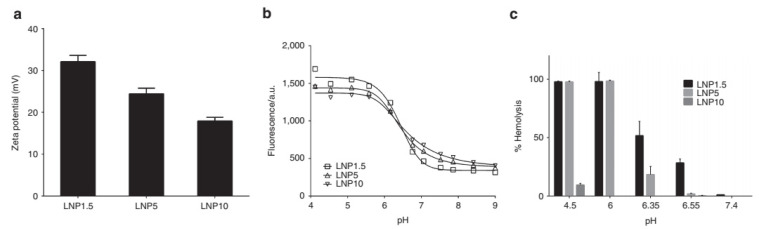
Increasing PEO density shielded the surface charge and reduces haemolytic activity on the LNPs. (**a**) Zeta potential measured at pH 5.5 for LNP1.5, LNP5, and LNP10 in RO/DI water. (**b**) The measurement of fluorescence (using 2-(p-toluidinyl) naphthalene-6-sulfonic acid sodium salts) at different pH range for LNP1.5, LNP5, and LNP10. The characteristic signal drop of fluorescent was observed which is attributed by pKa of IL (**c**) Haemolytic activity for LNP1.5, LNP5, or LNP10 across different pH ranges. The relative haemolysis is compared with complete haemolysis (100%) which is induced using Triton X-100 at 2% is shown. Adapted from Ref. [70], Mol. Ther. Nucleic Acids 2014.

**Figure 4 pharmaceutics-14-02520-f004:**
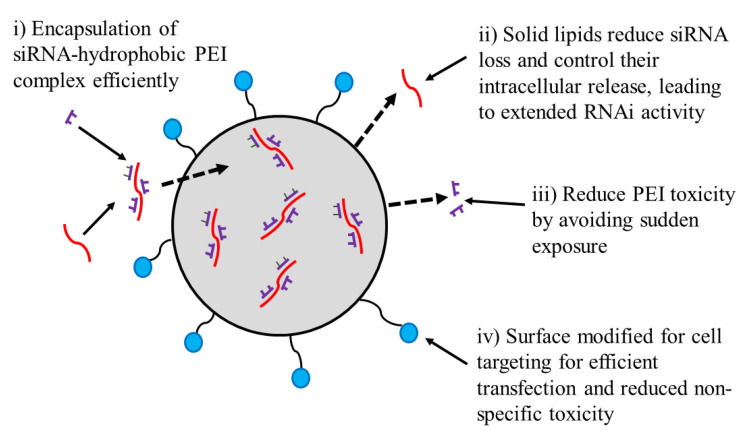
Schematic design and hypothesis of using LPNs for providing targeted, extended, and safer siRNA therapy. siRNA molecules (red) are complexed with hydrophobically modified linear PEI molecules (violet lines) which are then physically encapsulated into the core of a nanocarrier made up of solid lipids (grey). The surface groups represent the targeting moieties, e.g., phospholipids tagged with the folate group (blue).

**Figure 5 pharmaceutics-14-02520-f005:**
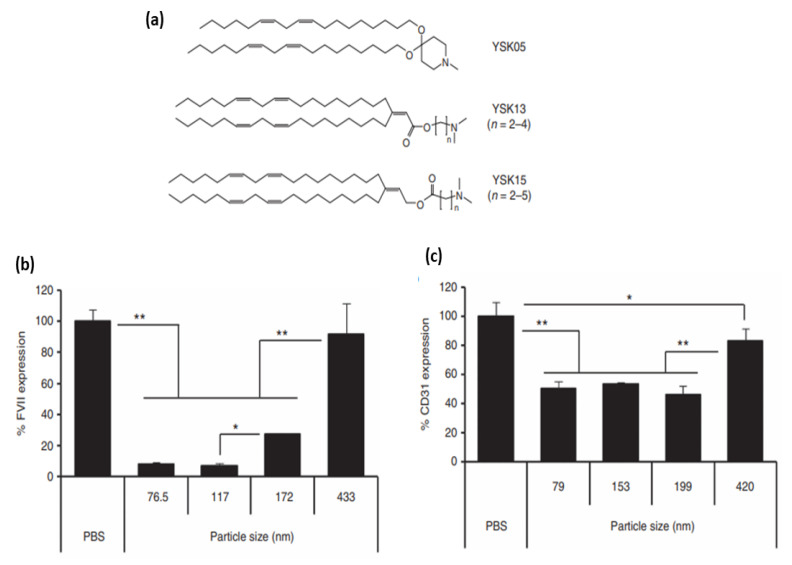
(**a**) Different cationic lipid structures. (**b**) Effect of size on the gene silencing activity in hepatocytes and LSECs. Mice were injected with siFVII formulated in various sizes of YSK13-C3-MENDs at a dose of 0.1 mg/kg and plasma FVII activities were measured in 24 h (**c**) Mice were injected with siCD31 formulated in various sizes of YSK15-C4-MENDs at a dose of 0.1 mg/kg and quantification of CD31 mRNA expression on LSECs in 24 h. Data are represented as the mean ± SD (*n* = 3). * *p* < 0.05, ** *p* < 0.01 (by one-way nrANOVA, followed by SNK test. nrANOVA, non-repeated analysis of variance; SNK, Student-Newman-Keuls. Adapted from Ref. [114], Mol. Ther. 2016.

**Figure 6 pharmaceutics-14-02520-f006:**
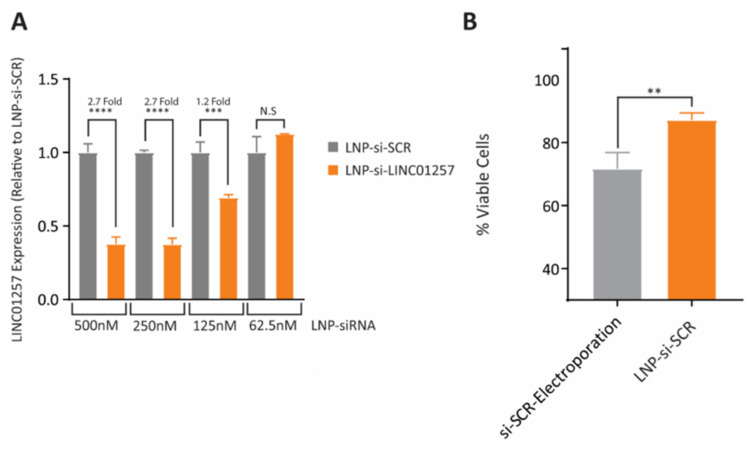
si-LINC01257 siRNA-loaded LNPs effectively target the expression of LINC01257 when tested in Kasumi-1 cells. (**A**) RT-qPCR profile of LINC01257 expression at varying concentrations of LNP-si-SCR after 72 h of treatment. (**B**) Viability of Kasumi-1 cells following either 72 h treatment with LNP-si-SCR or electroporation with si-SCR. **** = *p* < 0.0001, *** = *p* < 0.001, ** = *p* < 0.01 N.S = no significance, *n* = 3. Adapted from Ref. [165], Pharmaceutics 2021.

**Figure 7 pharmaceutics-14-02520-f007:**
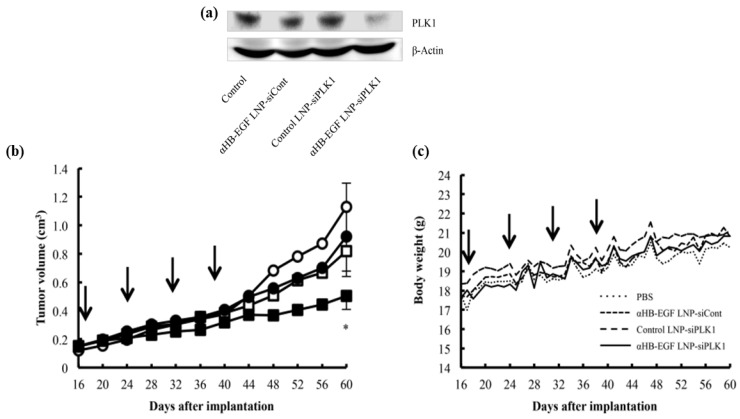
(**a**) Gene silencing activity of αHB-EGF LNP-siPLK1 in vivo study. MDAMB-231 carcinoma-bearing mice were injected via the tail vein with αHB-EGF LNPsi Cont, Control LNP-siPLK1, or αHB-EGF LNP-siPLK1. After treatment, the tumour was homogenized on the 5th day. Using the Western blotting technique, PLK1-protein expression was determined. (**b**) Anticancer effect of αHB-EGF LNP-siPLK1 in vivo. Once a week the mice with MDA-MB-231 carcinoma were injected four times with PBS (○), αHB-EGF LNP-siCont (□), Control LNP-siPLK1 (●), or αHB-EGF LNP-siPLK1 (■). The size of the tumour for each mouse was monitored from day 16. The days of treatment are indicated with arrows. The asterisk represents the significant difference (* *p* < 0.05 vs. PBS). (**c**) Monitored body weight of treated mice. Adapted with permission from Ref. [173], Mol. Pharm. 2018.

**Figure 8 pharmaceutics-14-02520-f008:**
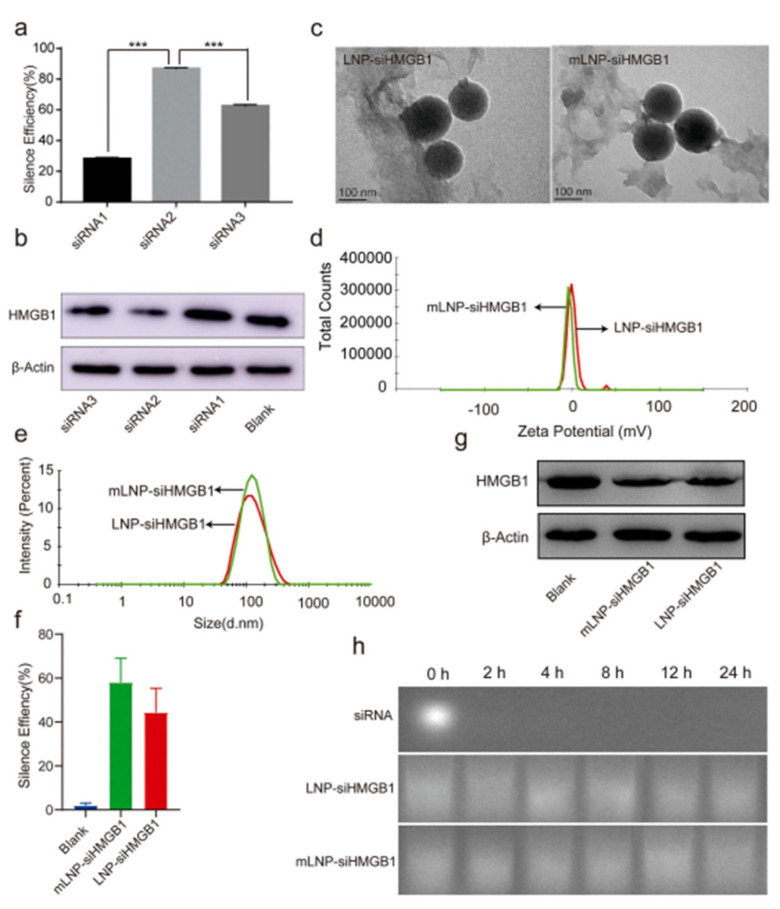
siRNA sequences screening and characterization of mLNP-siHMGB1. (**a**) Gene silencing efficiency (QPCR) of three siRNA sequences. More than 80% of silencing efficiency was achieved by the second siRNAi. *** represent *p* < 0.001. (**b**) Western blot image displaying the effect of three different siRNA sequences on the HMGB1 protein expression; suggesting that siRNA2 sequences significantly decreased the HMGB1 protein expression. (**c**) TEM images suggested the spheroidal structures of LNP-siHMGB1 and mLNP-siHMGB1. (**d**) Zeta potential is nearly 0 mV. (**e**) Mean particle size was around 100 nm, with <0.2 polydispersity index (PDI), (**f**) LNP-siHMGB1 and MLNP-siHMGB1silencing efficiency. (**g**) HMGB1 protein expression was reduced by LNP-siHMGB1 and mLNP-siHMGB1 treatment as indicated by Western blot analysis, and (**h**) The LNP or mLNP encapsulating the siHMGB1 did not deteriorate within 24 h in the serum, indicating good serum stability. Adapted with permission from Ref. [174], J. Control. Release 2022.

**Figure 9 pharmaceutics-14-02520-f009:**
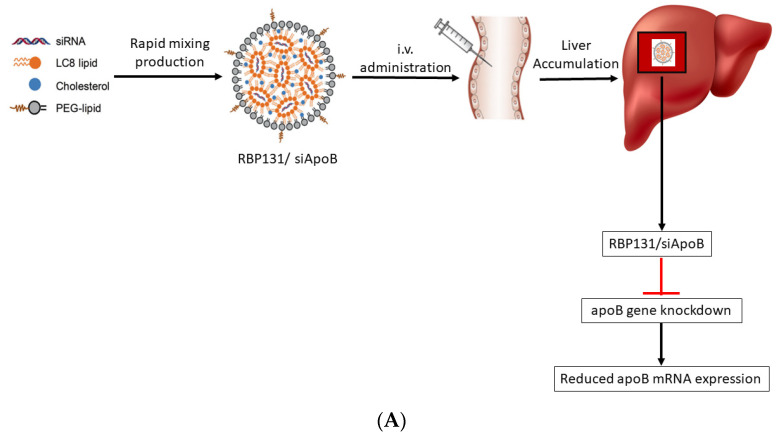
(**A**)**:** Schematic representation of RBP131/siApoB-mediated gene silencing. The siApoB encapsulated by RBP131 (RBP131/siApoB) was injected IV into C57BL/6 mice. After reaching the target site (liver), the RBP131/siApoB binds to the apoB gene and degrades it reducing the apoB protein levels. (**B**) Characteristics of RBP131/siRNA NPs. (**a**) Graphical representation of RBP131/siRNA NP and its compositions. (**b**) Cryo-EM image of RBP131/siRNA NPs. (**c**) Images displaying liquid and lyophilized forms of RBP131/siRNA formulations. (**d**) Particle size. (**e**) PDI. (**f**) Encapsulation efficiency of RBP131/siRNA kept at varying temperatures for different weeks, respectively. (**g**) Details of RBP131/siRNA formulation properties. (**h**) Graph displaying pKa titration of RBP131/siRNA complex. Adapted from Ref. [175], Signal Transduct. Target. Ther. 2022.

**Figure 10 pharmaceutics-14-02520-f010:**
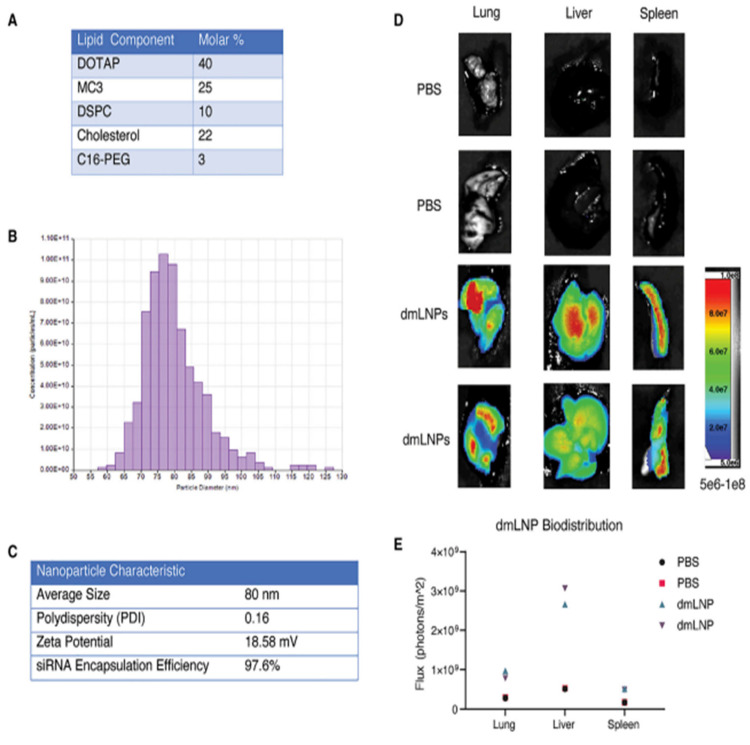
Characterization and biodistribution of dmLNP-siRNAs (**A**) dmLNP-siRNA LNPs molar composition. (**B**) Determination of size distribution of dmLNP-siRNA LNPs using the qNano Gold tuneable resistive pulse sensing device. (**C**) Characteristics of dmLNP-siRNA NP include size, zeta potential, PDI, and the encapsulation efficiency of siRNA. (**D**) Biodistribution of dmLNP-siRNA in C57/BL6 mice that received DiD-labeled dmLNP-siRNA NPs via retro-orbital (RO) route at a dose of 1 mg/kg siRNA or PBS vehicle as control. After 24 h injection, the liver, lung, and spleen were removed after the mice were sacrificed. Organs were imaged using a LagoX small animal imaging machine for DiD fluorescence at 640 nm excitation and 690 nm emission wavelength. (**E**) Quantitative analysis of DiD fluorescence with *n* = 2 mice per treatment group in each organ. Adapted from Ref. [188], Mol. Ther. 2021.

**Figure 11 pharmaceutics-14-02520-f011:**
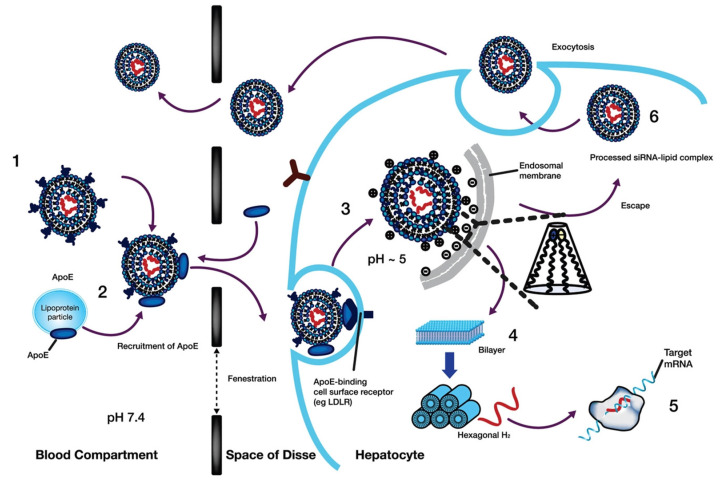
Proposed mechanism of action for Patisiran LNP. (1) Dissociation of PEG_2000_-C-DMG from LNP after IV administration. (2) Removal of the PEG followed by ApoE- dependent process, thereby facilitating uptake of LNP into hepatocytes. (3) Endocytosis-mediated internalization, ionization of DLin-MC3-DMA lipid as a result of decrease in pH in endosome. (4) Destabilization of endosomal membrane resulting from interaction of positively charged DLin-MC3-DMA lipid with the negatively charged endosomal membrane lipid, as a result LNP disintegrates, releasing ALN-18328 into the cytoplasm. (5) The released ALN-18328 binds to RISC, which then degrades the target TTR mRNA and subsequently reduced the levels of target protein. (6) Exocytosis of remaining proportion of LNPs, departure from late endosomes or lysosomes into the circulation. Adopted from Ref. [195], J. Clin. Pharmacol. 2020.

**Table 3 pharmaceutics-14-02520-t003:** Proposed siRNA molecules with GC%, free energy of binding, and score. Adapted from Ref. [190], Comput. Biol. Chem. 2021.

Target	Predicted Duplex siRNA Candidate at 37 °C	GC%	Free Energy of Binding (kcal/mol)	Score
Sequence 1	AGUAGAAAUACCAUCUUGGAC CCAAGAUGGUAUUUCUACUAC	38	−31.50	0.946
Sequence 2	UUUCUUAGUGACAGUUUGGCC CCAAACUGUCACUAAGAAAUC	40	−34.54	0.861
Sequence 3	ACAUUGUAUGCUUUAGUGGCA CCACUAAAGCAUACAAUGUAA	36	−30.74	0.986
Sequence 4	AAUUUGCGGCCAAUGUUUGUA CAAACAUUGGCCGCAAAUUGC	43	−31.61	0.793

## Data Availability

Not applicable.

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
