# Peer review of "siRNA Functionalized Lipid Nanoparticles (LNPs) in Management of Diseases"

_pharmaceutics, 2022, doi:10.3390/pharmaceutics14112520_

Round 1
Reviewer 1 Report
In this review article, the authors presented "siRNA functionalized Lipid Nanoparticles (LNPs) in management of cancer and other infectious diseases". They described all the different lipid-based nanoparticles, their synthesis methods, and their properties. They have also described the biological barriers for lipid nanoparticles which is a crucial aspect while designing the nanoparticles. Finally, they discussed the applications of lipid nanoparticles in various diseases. From my point of view, the topic is fascinating. The review paper is concise and well-written. I recommend the publication of this review article in Pharmaceutics after minor revisions. My comments are as follows:
1) The authors can modify the title of the review. It’s not necessary to specify the different diseases, such as cancer and infectious disease. They can generalize the title of the review and later on specify the various diseases.
2) The authors should highlight the novelty aspect of this review in the last paragraph of the introduction or maybe in the abstract, so readers know what makes this article different from the others available in the literature.
3) The authors may also elaborate on different types of LNPs. What are the different lipids available, which lipid is mostly used for different biological applications, and why? All these kinds of questions should be answered to clear the picture for the readers.
Author Response
Reviewer 1
The authors can modify the title of the review. It’s not necessary to specify the different diseases, such as cancer and infectious disease. They can generalize the title of the review and later on specify the various diseases.
- Thank you for the comments. The new title will be “siRNA functionalized Lipid Nanoparticles (LNPs) in management of diseases”.
The authors should highlight the novelty aspect of this review in the last paragraph of the introduction or maybe in the abstract, so readers know what makes this article different from the others available in the literature.
- As suggested, we have included few lines in the last paragraph of the introduction
The authors may also elaborate on different types of LNPs. What are the different lipids available, which lipid is mostly used for different biological applications, and why? All these kinds of questions should be answered to clear the picture for the readers
- Thank you for the comments. Regarding the different types of lipids available for LNPs we have already prepared a table mentioning the different lipids commonly used for preparation of LNP (Table 1)
- Based upon the required application, numerous different lipids are available and have been widely used by researchers. For the synthesis of each LNPs (liposomes/Solid lipid Nanoparticle/ Nanostructured Lipid Carrier/ Nanoemulsions), different lipids are required. For instance, for liposome synthesis, different phospholipids such as, PE, PG, PS and PC, cholesterol, PEG lipids are required, like wise for SLN, solid lipids such as stearic acid, palmitic acid, surfactants such as Tween 20, 80, Span 40, 80 are used. As such, the requirement of lipids solely depends upon the type of LNP and application
Reviewer 2 Report
The review entitled ‘’ describes the role of siRNA based Lipid Nanoparticles (LNPs) in cancer and infectious diseases. This review is well written and it will be helpful for readers working in this field. Below are some suggestions for the authors which can improve the manuscript.
1. Authors should cite the references in the table number 1.
2. Authors should propose the mechanism of action of siRNA based Lipid Nanoparticles in different types of cancer. There should be systematic diagram or figure to support the hypothesis. This will provide a clear insight for the readers.
3. In the title infectious diseases are mentioned this is a generalized term authors should modify the title as in the text only few infectious diseases has been discussed. The authors should focus on few infectious diseases and describe in detail the mode of action of these LNPs.
4. There should be a separate section for the methods of designing different siRNA as there are lot of tools available. This should be discussed in details.
5. manuscript should be accepted after revision.
Author Response
Authors should cite the references in the table number 1.
- As per suggested, 5 references have been added in Table no 1.
Authors should propose the mechanism of action of siRNA based Lipid Nanoparticles in different types of cancer. There should be systematic diagram or figure to support the hypothesis. This will provide a clear insight for the readers.
- Thank you for the comments. A schematic figure (Figure 1) describing the mechanism of action of siRNA-based Lipid Nanoparticles has been added. In addition new Figures 9 and 11 provide info on this issue.
In the title infectious diseases are mentioned this is a generalized term authors should modify the title as in the text only few infectious diseases has been discussed.
- Thank you for the comments. The new title will be “siRNA functionalized Lipid Nanoparticles (LNPs) in management of diseases’’
The authors should focus on few infectious diseases and describe in detail the mode of action of these LNPs.
- As per suggested, a schematic figures (Figure 9a and 11) explaining the mode of action of lipid nanoparticles and siRNA based gene silencing has been added.
There should be a separate section for the methods of designing different siRNA as there are lot of tools available. This should be discussed in details.
- We did not take any action on this comment. We dispersed this knowledge at different points in the manuscript and prefer not to combine them into a single separate section. We are not experts in this area (hence cannot do an in-depth analysis of the field) and only referred to computational studies that are published by others. Hope this is acceptable to the reviewer.
Round 2
Reviewer 2 Report
Authors have addressed the comments thus manuscript should be accepted